# Molecular Imprinted Polymers Coupled to Photonic Structures in Biosensors: The State of Art

**DOI:** 10.3390/s20185069

**Published:** 2020-09-07

**Authors:** Andrea Chiappini, Laura Pasquardini, Alessandra Maria Bossi

**Affiliations:** 1Institute of Photonics and Nanotechnologies (IFN-CNR) CSMFO Laboratory and Fondazione Bruno Kessler (FBK) Photonics Unit, via alla Cascata 56/C, 38123 Povo Trento, Italy; andrea.chiappini@ifn.cnr.it; 2Indivenire Srl, via Alla Cascata 56/C, 38123 Povo Trento, Italy; l.pasquardini@indiveni.re; 3Department of Biotechnology, University of Verona, Cà Vignal 1, Strada Le Grazie 15, 37134 Verona, Italy

**Keywords:** molecularly imprinted polymers, photonic structures, bio/sensors, surface plasmon resonance, photonic crystals, optical fiber, optical waveguide

## Abstract

Optical sensing, taking advantage of the variety of available optical structures, is a rapidly expanding area. Over recent years, whispering gallery mode resonators, photonic crystals, optical waveguides, optical fibers and surface plasmon resonance have been exploited to devise different optical sensing configurations. In the present review, we report on the state of the art of optical sensing devices based on the aforementioned optical structures and on synthetic receptors prepared by means of the molecular imprinting technology. Molecularly imprinted polymers (MIPs) are polymeric receptors, cheap and robust, with high affinity and selectivity, prepared by a template assisted synthesis. The state of the art of the MIP functionalized optical structures is critically discussed, highlighting the key progresses that enabled the achievement of improved sensing performances, the merits and the limits both in MIP synthetic strategies and in MIP coupling.

## 1. Introduction

Optical sensors comply with the definition of self-contained devices, consisting of a molecular recognition element (MRE), apt to the selective interaction with a target analyte, and of an optical transducer element, that converts the molecular interactions into optoelectronic signals, providing quali-quantitative analytical information about the targeted physical or chemical species.

In general, optical sensors are characterized by fast to real-time responses, by the direct, or label free, detection and by the ability to monitor target analytes in complex environments. Moreover, optical sensors withstand the criteria of robustness, miniaturization and the possibility to operate in remote mode [1,2]. The market for commercialized sensors has been growing with a positive trend and at a steady pace [3,4,5].

Taking advantage of the wide variety of optical structures, each characterized by a specific pattern and by unique features, such as interferometers, optical waveguides, optical resonators, optical fibers and optical sensing can rely on a number of different configurations. Furthermore, when optical sensors are based on plasmonics phenomena [6], that provide the ability for extreme light control, additional analytical advantages, such as determinations down to the single molecule, in vivo and in situ can be attained [7].

The present review discusses the advancements in optical sensors, classified according to the type of optical structure, with particular emphasis on the recent literature, focusing specifically on a particular kind of MRE that is a molecularly imprinted polymer (MIP).

MIPs are polymeric synthetic receptors prepared by means of a template assisted synthesis. The use of MIPs in optical sensing offers a number of advantages, such as the possibility to prepare an MIP-MRE suitable to capture any kind of target analyte, from small molecules to proteins [8], cells and viruses [9]. MIPs can be addressed at the recognition of immunogenic and toxic analytes, and can be entailed to the desired degree of affinity and selectivity [10,11], so that some of the MIP materials demonstrated to outpass the characteristics of their natural counterparts, such as antibodies or biological receptors. Moreover, MIPs are robust, plastic-based materials, cheap to synthesize and prone to be reused. MIPs are suitable to work in aqueous as well as in organics, stable over time and in extreme conditions (over the whole pH scale; reported working temperatures range from 0 to 150 °C [12] and suitable for sterilization) [13]. Concerning the physical properties and the appearance of the MIP materials, MIPs can be prepared from a wide choice of protocols that enable these materials to be obtained in many different formats, from macro- to micro- and nanosizes, from thick to thin –sub-nanometer-layers and from micro- to nanoparticles; moreover, MIPs can be integrated to devices and electronics by means of easy and straightforward chemical-coupling procedures. Hence, the versatility of the MIP materials appears to be ideal to impart the desired selectivity to the various optical structures, opening up the possibility to explore new sensor configurations. In fact, a number of commercial enterprises produce MIPs for sensors [14,15,16].

## 2. Molecularly Imprinted Polymers (MIPs)

MIPs are a class of polymeric receptors whose recognition properties are entailed during the very synthetic process [17,18]. The process of molecular imprinting, as shown in Figure 1, consists of solvating the target molecule to be printed, called the template, together with monomers, capable of forming either weak noncovalent [17], or reversible covalent [18], interactions with the template. The so-formed prepolymerization complex is made up of crosslinkers, of the catalyst and is polymerized. At the end of the polymerization, the MIP material is washed extensively, so to extract the templates. The resulting MIP material possesses embedded binding cavities that are complementary in shape, and in the sterical placement of the chemical functionalities, to the template; hence, the cavities are suitable to selectively rebind it. In fact, MIPs have been demonstrated to bind the template molecule displaying selectivity and specificity alike, if not surpassing natural receptors and antibodies; the selective binding occurs even if the target template is in a complex matrix, such as a biological fluid [19,20], or if the target template is on the surface of a cell or in a tissue [21,22,23,24].

Technical progresses in mastering the polymer synthesis [25] have permitted the leap forward in the MIP preparation to be made, so that nowadays MIPs can be prepared in different formats, ranging from macro-, to micro- and nanomaterials [26].

It is worth nothing that MIP advances in the formation of layers of controlled thickness and softness, and methods for the polymerization or deposition in situ permit an on-demand modification of the optical structures with the MIP MRE, optimizing the quantity of selective binding sites and the optical properties of the MIP MRE. For this purpose, initially, the MIPs were dropped on the optical sensor and chemically- or photopolymerized in situ, forming micrometer thick layers of MIP. Control over the MIP thickness was introduced by spin-coating prepolymerization mixtures solvated in various porogenic solvents characterized by different densities, so as to finally modulate the mesostructure of the formed MIPs and the thickness of the MIP layer [27].

Later, a high degree of control over the MIP layer was achieved by introducing the use of micro- and nanolithography, which permitted to deposit precise micro- and nanodots [28,29]. Moreover, further improvements were enabled by the use of in situ light-induced photopolymerization strategies, such as evanescent wave photopolymerization to prepare ultrathin MIP microdots directly on the optical structures [30]. As an example, the evanescent field demonstrated the “click” of the polymerization on the surface of an optical waveguide and was used to fabricate a composite optical fiber taper probe for sensing. Achieving a high control of the polymerization permitted the polymerization within the scale of the evanescent field on the surface of the fiber to be effectively maintained to produce MIP nanometric structures [31]. Moreover, also two-photon stereolithography (TPS) was successfully developed for preparing MIPs, choosing a suitable photoinitiator and compatible MIP formulations so that the femtosecond laser irradiation for TPS preserved the imprinting effect and allowed high-resolution direct-writing in 3D of the MIPs [32].

Alternatively, the control over the MIP deposition and/or decoration of the optical sensing surfaces can be achieved by following a different conceptual path, which is exploiting methods to prepare of MIP micro and nanostructures later integrated in the optical sensor.

The optimization of polymerization methods such as precipitation polymerization [33], solid phase synthesis [34], directional syntheses [35,36] allowed to scale down the MIPs to the nanometric dimension (20–200 nm) [37].

The nanometric MIP particulate, called MIP NPs, displays an interesting close resemblance to the antibodies: low number of binding sites per nanoparticle [38], fast kinetics of interaction and high affinities (dissociation constants K_D_ ≤ 10^−9^ M), attuned selectivity, while sharing the robustness and shelf-lives of polymers; hence, the MIP NPs can represent ideal “plastic antibodies” to be integrated in sensors [39].

The MIP composition can be optimized by means of molecular modeling tools [29], while in terms of materials, the most exploited for the preparation of MIP layers and nanoparticles are polyacrylates, polymethacrylates, polyacrylamides and electroactive materials such as anilines and p-phenylboronates [40]. Of particular interest is the class of nanogels, such as poly-N-isopropylacrylamide and copolymers [34,35], for their responsive properties of shrinking and swelling dependent on the environment [41] that offer advantages in terms of adaptive recognition sites, thus making MIP NPs suitable for the recognition of macromolecules [42] and for signal amplification [41].

Regarding the issue of the integration of the MIP NPs to the optical transducer element, the MIP NPs can be coupled to the optical sensing structure by the immobilization chemistries that are state of the art in biomacromolecular conjugation [43].

## 3. Photonic Structures

In recent years, outstanding progress has been reached in the field of optical platforms, which have been leading to optimized sensing performances [44]. Moreover, the availability of reliable optical platforms with good mechanical stability, reproducibility and, last but not least, user-friendliness has been foreground for the integration of sensors in lab-on-chip systems, promoting the development of fully automated systems with practical impact on both our present daily life and our lives in the near future.

In this section, we will introduce the main optical platforms exploited for the development of optical sensors addressing their working principles, properties, characteristics and commercialization. The optical platforms presented in the next section are based on surface plasmon resonance (SPR), whispering gallery mode (WGM), waveguides and photonic crystals (PC). Each platform is schematically depicted in Figure 2.

### 3.1. Surface Plasmon Resonance

Surface Plasmon Resonance (SPR) is currently the most widespread and well-known transduction method for the development of bio/sensors. A surface plasmon (SP) wave can be optically induced on the metal–dielectric interface selecting a wave vector that matches the one of the SP mode. SPR is based on a change in the refractive index (*n*) caused by molecular interactions occurring close to the metal surface and detected via the surface plasmon (SP) wave. The molecular interaction induces a shift in the position of the SPR peak associated to the variation of the refractive index resulting in a change in the SP wave vector (*k_sp_*). From a mathematical point of view, the wave vector can be expressed by the following expression (Equation (1)) [6]:(1)$$k_{sp}\;=\;\frac{2\pi }{\lambda }\cdot \sqrt{\frac{n_{m}^{2}\cdot n_{s}^{2}}{n_{m}^{2}+n_{s}^{2}}}$$
where *n_m_* is the refractive index of the metal and *n_s_* is the refractive index of the sample. Any change in the surrounding environment, or in the material, placed at the interface between the metal and dielectric, such as the binding of the target analyte to the MRE (see Figure 2a) alters the momentum of the SP (*k_sp_*); as a consequence, SPR no longer occurs at the previous incidence angle, and an SPR shift takes place. Such shift in the resonance angle is directly proportional to the change in the refractive index which can hence be quantitatively correlated to the presence of the analyte. SPR platforms can be considered the most mature ones in terms of technology, so that nowadays, a number of commercial SPR sensors is present in the market and available from companies including Biacore, IBIS Technologies B.V., AutoLab, GWC Technologies, Genoptics Bio Interactions, Biosensing Instrument and SPR Navi [49].

### 3.2. Whispering Gallery Modes Resonators

An interesting photonic structure for the development of highly sensitive bio/sensors is represented by whispering gallery mode resonators (WGMR). This exploits the circular symmetry of the resonators to sustain the so-called whispering gallery modes, interpretable as circulating acoustic or electromagnetic waves strongly confined within the structure, as shown in Figure 2b. When the refractive index of a resonator (such as microsphere, microring, microdisk) is larger than that of the ambient medium and the incident angle exceeds the critical angle of the total internal reflection, light is confined in the cavity, as evidenced in Figure 2b. Moreover, in WGMR structures, the electromagnetic waves are confined in a very tiny volume and present extremely small mode volume, with very high power density and very narrow spectral linewidth. From a mathematical point of view, the wavelength’s light trapped inside the WGM can be expressed by the following equation:(2)$$2\pi R_{WGM}\;=\;m\frac{\lambda }{n_{eff}}$$
where *R_WGM_* corresponds to the radius of the cavity, *λ* is the resonance wavelength, *n_eff_* is the effective mode index of the WGM and *m* denotes the polar order of the WGM. The peculiar features of WGMR have been exploited to reach ultrahigh performances for chemical, biochemical and gas sensing and even to detect single molecules [50]. The readout working principle in WGMR sensors is based on a frequency shift (Δ*λ*) of the resonance induced by an external stimulus (analyte, gas, etc.) interacting with the resonator and produces a modification in the refractive index and/or dimensions of the resonator, as described by the Equation (3).
(3)$$\frac{\Delta \lambda }{\lambda_{0}}\;=\;\frac{\Delta R_{0}}{R_{0}}+\frac{\Delta N_{s}}{N_{s}}$$
where *R*_0_ is the radius of the resonator and *N_s_* is its refractive index.

Different shapes such as spheres, bubbles, bottles, disks, toroids, rings, capillaries and cylinders are available, and different inorganic materials such as glass, quartz, silicon, silicon-on-insulator (SOI), silicon nitride and crystals or organic materials such as polymers can be used [50].

Although this kind of structures is widely employed in the experimental laboratories, in order to be really present in the market, some issues must be addressed, in particular the stability and the reproducibility of the coupling have to be guaranteed [51]. Concerning the last point, it is worth mentioning that the choice of the material is crucial since material losses, such as absorption and surface scattering, play a key role in determining mode linewidth hence affecting the final sensitivity of the sensor. Moreover, materials that present large thermo-optic and thermal expansion coefficients are to be preferred in order to assure the thermal stability. Although this platform permits the development of highly sensitive sensors, it is worth mentioning that expensive apparatus, such as narrow line tunable lasers and optical spectrum analyzers, are required both for the interrogation and for system detection.

### 3.3. Optical Waveguide Lightmode Spectroscopy

Optical Waveguide Lightmode Spectroscopy (OWLS) is widely applied to monitor protein adsorption, polymer self-assembly and living cells on the surface of the sensor in a label-free manner [52], but also environmental and health aspects have been addressed. In this approach, as shown in Figure 2c, the optical element is an optical waveguide (both planar and fiber-optic) in which the light is confined in the medium that presents high permittivity/high refractive index surrounded on all sides, by materials with lower refractive indices, such as a substrate and the media to be sensed.

Over recent years, exploiting OWLS; several transduction schemes such as fluorescence, interferometry, polarization; and evanescent-based detection have been proposed and coupled to waveguides and fiber optics [53]. In these configurations, an optical wave propagates inside the waveguiding film and its evanescent tail penetrates into the cladding layer, which is placed directly in the bulk solution or is covered by a layer suitable to recognize the target molecule. Variation in the local refractive index within the evanescent field tail produces a consequent modification of the optical response, or an emission in the case of a fluorescence-based approach.

OWLS techniques correlate the variation of the propagation intensity, phase or the emission to the concentration of the analytes to be detected.

The two main configurations exploiting waveguides are the so-called: (a) slot-waveguide-based biosensors and (b) interferometric-waveguide-based biosensors. The first system is formed by two slab waveguides with high refractive indices situated very close to one another, such as a gap with a medium having a low refractive index (air) is formed. The interaction of the analyte with these structures produces a variation of the refractive index causing a shift in the wavelength of the propagating mode, thus the resonance shifts reveal the analyte concentration.

The interferometric waveguides consist of two couplers (or Y branches) connected by two waveguides: one acting as a reference port and the second one as the sensitive element. The analyte interaction induces a change in the refractive index, producing phase-shifts in the mode. By combining the phase-shifted mode with the reference one, an interference signal can be generated that is suitable for monitoring the phase difference as sensor output, ultimately allowing the quantitative determination of the analyte.

Crucial is the kind of materials used for the OWLS, the dimensions of the components of the waveguide and their effect on the evanescent field. For example, a waveguide with a small size (of the order of hundreds of nanometers) and a large refractive index difference (Δn) between the guiding and cladding films (Δn > 2 @ 633 nm) provides an intense evanescent field and, therefore, a better coupling condition with the analyte. From an applicative point of view, OWLS can be considered a mature and versatile approach: SME companies such as Lead Pharma, LioniX and Surfix take advantage of this technique to develop detection platforms for biologicals, such as proteins and viruses. However, it is worth mentioning that the realization of these systems needs the employment of lithography and selective etching apparatus for the definition of the suitable geometry (i.e., slot-waveguide-based biosensors or interferometric-waveguide-based biosensors) [44].

### 3.4. Photonic Crystals/Colloidal Crystals

The last optical structure discussed in the present review relates to photonic crystals (PC). PC structures have been extensively applied in microfluidics, telemedicine, smart materials and flexible sensors. In particular, over recent years, colloidal crystals have been employed as suitable optical platforms for the development of chemical, physical and biosensors based on a chromatic response under an external stimulus [54]. As an example, Figure 2d depicts a PC composed by a periodic arrangement of regularly shaped elements with different dielectric constants; the periodicity in the dielectric materials produces a photonic band gap (PBG), so that certain wavelengths, due to the presence of the PBG, cannot propagate but is reflected. Moreover, by properly choosing the PC materials and the dimensions of the PBG-constituting elements, it is possible to design a PC that present an initial response in the visible region (see Figure 2d). Now, when an external stimulus interacts with the PC platform, this induces a change in the physical features of the PC producing a color change that can be associated to the concentration/magnitude of the external stimulus. The variation in the wavelength position of the reflected light can be described by Bragg’s and Snell’s laws (reported in Equation (4)).
(4)$$m\cdot \lambda \;=\;2\cdot d\cdot n_{eff}$$
where *m* is the order of the reflection, *n_eff_* is the effective refractive index, *d* is the lattice constant and *λ* is wavelength of the reflected light.

According to the above mentioned law, the reflection wavelength of PCs depends on the refractive index contrast between two periodic media (*n_eff_*) and the lattice constant (*d*). Consequently, the absorption or the immobilization of the analyte induces a variation in the refractive index (RI), since the RI of the target molecule is different from that of the PC host matrix. On the other hand, swelling or shrinking upon binding the target molecule, leads to changes of the lattice constant producing a shift in the reflected light (chromatic variation) as evidenced in Figure 2d. Looking at Equation (4), it becomes evident that choosing, in an appropriate way, the kind of material and its dimensions opens the possibility to develop a chromatic sensor whose response is designed to occur in the visible range.

The key advantages of such kinds of structures are the detection based on a visual response of the sensor, potentially avoiding any signal transduction; hence, this characteristic could favor the diffusion of these systems as simple and safe devices usable by untrained end-users in different applications fields. Despite the potential, only few commercial devices are present at the moment on the market, mainly purchased by Opalux Inc. [55], probably due to the fact that the realization of this kind of system is highly time consuming.

## 4. MIP Coupled to Photonic Structures

### 4.1. MIP and SPR

Surface plasmon resonance (SPR) is the earliest optical mode functionalized with MIPs for sensing and biosensing [56]. Later, it became the most exploited one, counting nowadays several hundreds of papers (source: Scopus database; SPR & MIP n. 263 original papers at August 2020). The following section discusses the key strategies to devise MIP-SPR and their impact on the overall sensor performance.

#### 4.1.1. Deposition of MIP Layers

A key point in MIP-SPR sensing is the MIP functionalization of the optical platform; thus, efforts and progress has been dedicated to this crucial aspect. The easiest strategy to prepare an MIP-SPR structure is to deposit the MIP onto the chosen plasmonic platform (Table 1). The deposition followed by spin coating of the MIP and polymerization was reported by Devanathan and coworkers [57]. Authors formed an ultrathin—subnanometric—layer of MIP, prepared for the recognition of the δ-opioid G-protein, which offered ultralow sensitivities, high selectivity and subpicomolar binding affinity for the target protein, demonstrating great performances. Later, the same spin coating deposition process was exploited for SPR configurations based on optical fibers. Both MIP plastic optical fiber and an MIP silica optical fiber sensors were described, yet showing different performances [58,59,60]. In the case of the detection of profenofos [60], the dynamic range of concentration was from 10^−4^ to 10^−1^ μg/L, with a red shift of 18.7 nm in resonance wavelength over the whole concentration range, while the reported Limit of Detection (LoD) was of 2.5 × 10^−6^ μg/L.

The main weaknesses of the functionalization based on the MIP prepolymer deposition and spin coating, followed by in situ polymerization, are the reproducibility of the layer and the use of solvents to modulate the resulting MIP film thickness [27] that can be incompatible with the template solubility or with its integrity (e.g., solvents unfold proteins).

#### 4.1.2. MIP Grafting and In Situ Controlled Polymerization

A promising solution was to graft the MIP to the SPR platform, thus exploiting bottom up synthesis strategies (for a comparison, see Table 1). Lotierzo and colleagues proposed the functionalization of the gold surface with a self-assembled monolayer of 2-mercaptoethylamine onto which the photoinitiator 4,4′-azobis(cyanovaleric acid) was covalently coupled. An MIP thin film, composed of 2-(diethylamino) ethyl methacrylate and EGDMA was prepared, and a thickness of 40 nm was reported [61]. The group of Wei reported the MIP film synthesis in situ by UV photo polymerization on an SPR gold surface modified with 1-dodecanethiol and photoinitiator. Having testosterone as the target analyte, the increased homogeneity of the MIP layer allowed the sensor to reach a dynamic concentrations range of 1 × 10^−12^–1 × 10^−8^ mol/L, and an LoD of 10^−12^ mol/L [62].

Focusing on achieving a precise tuning of the MIP film thickness, controlled polymerization methods were applied to the in situ synthesis of MIPs. The group of Wei [63] studied the effects of reversible addition-fragmentation chain transfer (RAFT) polymerization on the preparation of the MIP-SPR platform to detect the hormone 17β-estradiol (E2). The photo-initiation was controlled by 2-methyl-2-[(dodecylsulfanylthiocarbonyl)sulfanyl]propanoic acid (DDMAT) in the absence of additional photoinitiators or catalysts. A homogeneous and reproducible MIP film permitted the sensor to respond to E2 in the range from 10^−14^ to 10^−6^ mol/L. The LoD was 1.15 × 10^−15^ mol/L (S/N = 3); moreover, the sensor was suitable for the detection of E2 in samples of tap waters. The MIP-SPR sensor was characterized by good reusability and stability (84% of the original response after 45 days of storage under dry and ambient conditions). Later, the same group reported on the use of a different chain transfer reagent, 2-methyl-2-[(dodecylsulfanylthiocarbonyl)sulfanyl]propanoic acid (TTCA), that was used to photopolymerize an MIP, composed of a dicarboxylic acid functional monomer (i.e., itaconic acid) and EGDMA, for the detection of progesterone. The MIP SPR sensor displayed excellent selectivity and was reused up to 8 adsorption-desorption cycles. Within the concentration range of 1 × 10^−18^ to 1 × 10^−8^ mol/L, the coupling angle change of SPR versus the negative logarithm of concentration showed excellent linearity: R^2^ = 0.99. The LoD was 0.28 × 10^−19^ mol/L. Finally, the sensor successfully analyzed progesterone in real samples [64].

Moreover, atom transfer radical polymerization (ATPR) was also exploited for the preparation of MIPs on the SPR platforms: MIP nanocavities for glycoproteins were fabricated via a bottom-up molecular imprinting approach using surface-initiated ATPR and the model glycoprotein ovalbumin, which was immobilized in a specific orientation onto an SPR chip by forming a conventional cyclic diester between the glycomoiety and a boronic acid and cis diol. The sensor reported selectivity for ovalbumin recognition, with an LoD of 6.41 ng/mL [65]. Hence, the reported sensors performances account for the effectiveness of strategies aimed at a fine control of the MIP polymerization.

As a last strategy, we mention the use of living radical polymerization for the preparation of MIP-SPR sensors, reported by Kidakova and colleagues [66]. The MIP was addressed at the selective recognition of a protein and was prepared by means of the synergistic use of the surface-initiated controlled/living radical photopolymerization and microcontact imprinting approach. An MIP-SPR sensor with high affinity (K_D_ in the pM range) and selectivity was obtained—confirming the key role played by the polymerization process of the MIP material on the metal surface—on the sensor performance.

#### 4.1.3. MIP Films by Lithographic and Printing Techniques

An alternative to the bottom up approaches that enable the growth of the MIP from the metal surface was to exploit the method for a highly controlled deposition of prepolymerized MIP solutions. Following this idea, matrix assisted pulsed laser evaporation was used to deposit MIP films of an amphiphilic block copolymer imprinted with an amino acid. The method avoided the need for a common solvent for the template and permitted the fabrication of layers with controlled thicknesses in the nanometer range [67]. Authors proved that the principle, despite the sensor response, was far from optimal. The group of Gyurcsányi proposed the use of standard photolithography to generate micropatterned surface-imprinted polymers (SIPs) for protein recognition. Avidin-imprinted poly(3,4-ethylenedioxythiophene)/poly(styrenesulfonate) (PEDOT/PSS) conducting polymer microbands were prepared on the SPR platform. The target analyte, avidin, was bound to the SIP with dissociation constants in the submicromolar range (125 nM), and the sensor demonstrated selectivity among functional homologues of avidin, i.e., neutravidin, extravidin and streptavidin [68]. A microcontact printing technique was proposed by the group of Denizli to produce MIP-SPR for procalcitonin with an LoD of 9.9 ng/mL [69] and for the detection of the bacterium monitoring Salmonella paratyphi in food supplies or contaminated water [70].

#### 4.1.4. Electropolymerization of MIP Films

Besides the functionalization of the SPR platform with acrylic, methacrylic and acrylamido based functional monomers and radical polymerizations, attempts were proposed to prepare controlled MIP films by using electropolymerization techniques and the aromatic monomers commonly exploited by electrochemical sensing. The idea was successfully demonstrated by Yu and Lai, using polypyrrole as a functional monomer and ochratoxin A as a target analyte [71]. Then, Advincula and coworkers proposed terthiophene and derivatives as suitable monomers for the electropolymerization of highly selective and robust ultrathin film MIP for SPR sensors, as in an example of detection of theophylline [72]. Alternatively, the same authors electropolymerized terthiophene together with carbazole electroactive monomers, demonstrating the possibility to expand the variety of available functional monomer combinations [73]. Finally, the same authors also showed that p-aminostryrene is suitable to form electropolymerized MIP thin films in an example addressed at sensing dopamine, reaching a remarkable picomolar detection of the analyte [74].

Gupta and coworkers proposed the electropolymerization of 3-aminophenylboronic acid (3-APBA) to nanopattern the SPR optical structure with cavities for the recognition of the T-2 toxin (T-2). Results showed that the MIP sensor exhibited a linear response for T-2 from 2.1 fM to 33.6 fM and had an LoD of 0.1 fM (0.05 pg/mL) [75]. Lately, Baldoneschi and coworkers demonstrated the use of norepinephrine as a new monomer for MIP electropolymerization [76]. Moreover, polynorepinephrine MIP films on SPR platforms showed a reduction in nonspecific binding, with respect to the classical polydopamine electropolymerized MIP films, and enabled the detection of the Troponin C biomarker for heart failure in the sub-nM range.

#### 4.1.5. Enhancing the Sensor Signal: LSPR and Responsive MIPs

The intensity of the signal arising at the binding of the analyte to the MIP-SPR platform plays a central role in the final sensor sensitivity, thus strategies to improve and enhance the signal have been devised. Among these, a pioneer is the work of Matsui and colleagues that placed an MIP hydrogel (acrylic acid, N-isopropylacrylamide, N,N′-methylenebisacrylamide) with embedded gold nanoparticles on the SPR sensing surface and showed that the swelling of the MIP hydrogel, triggered by the analyte binding events, caused distancing of the gold nanoparticles, shifting the dip of the SPR curve to a higher SPR angle, thus effectively enhancing the signal [77]. The idea was exploited in different applications (Table 1), including as an effective means to lower the LoD of *Escherichia coli* detection in urinary tract infections [78]. Additionally, the combination of metal nanostructures, such as NPs, and MIPs or of metals and MIP nanostructures, such as MIP NPs, have been studied for their ability to enhance the signal intensity. The group of Takeuchi reports a slab-type optical waveguide (s-OWG)-based microfluidic SPR measurement system for the detection of bisphenol A (BPA), that was based on MIP-NPs-immobilized consecutive parallel gold and silver deposition bands coexistent with BPA-AuNPs [79]. Later, the same group prepared a highly selective and sensitive nanosensing system for small molecules, based on the supraparticles of MIP-NPs and modified gold nanoparticles (Au-NPs). The MIP-NPs function was molecular recognition materials, while the Au-NPs function was signal transduction. The binding events produced spectral changes of LSPR of Au-NPs in the visible region, detected by means of a UV-vis spectrophotometer. Moreover, an affinity constant was obtained with this sensing system that was 120000 times larger than what previously reported for bulk MIP (respectively, 2.1 × 10^10^ M^−1^ and 1.72 × 10^5^ M^−1^). As a result, a sub-nM concentration of target molecules was successfully detected [80].

Au-NPs hybrid MIP-microgels proved to be suitable to monitor glucose levels, acting like a “glucose-indicator” in tear fluids. A visually evident color shift, from yellow to red, of the hybrid MIP microgel dispersion in response to glucose, over a clinically relevant glucose concentration range of 0.1–20 mM (1.8–360 mg/dL), was visible without instrumental aid [81].

Along the line of coupling responsive MIPs and metals, magnetic MIP NPs were exploited for amplifying SPR response. The magnetic MIP was designed by self-polymerization of dopamine on the Fe_3_O_4_ NPs surface in weak base aqueous solution in the presence of template chlorpyrifos. The sensor response was linear for the detection of the analyte over the range of concentrations from 0.001 to 10 μM with an LoD of 0.76 nM, proving excellent signal enhancement [82]. In another example, Cennamo and coworkers showed that an MIP polymer for the detection of TNT with embedded nanostars permitted the achievement of a sensitivity of 8.5 × 10^4^ nm/M—three times higher than in the case of the gold layer sensor [83]. MIP NPs, core-shell nanocomposite NPs (Ag@AuNPs), which incorporated hexagonal boron nitride (HBN) nanosheets and MIP (poly(2-hydroxyethyl methacrylate-methacryloylamidoglutamic acid), were prepared for etoposide detection. The results showed that the linear detection range was 1.70 × 10^−12^–1.70 × 10^−9^ M, and the LoD was 4.25 × 10^−13^ M [84].

Yet, exploiting metal nanostructures for signal enhancement is fine, but in principle, signal enhancement can also be obtained thanks to the use of soft, deformable or reactive MIP polymers, both in the form of layers and NPs (Table 1).

In fact, in an early report, poly-N-(N-propyl)acrylamide particles of about 300 nm in diameter, designed to swell and shrink as a function of analyte concentration in aqueous media, and spin coated onto a gold surface, showed a response in the µM concentration for theophylline [85]. Later, Zhou reported on the preparation of an MIP hydrogel film on the gold substrate containing a minute amount of cross-linker, so as to form loose gel structures for the fabrication of an MIP-SPR sensor for 3,3′-dichlorobenzidine. The sensing of 3,3′-dichlorobenzidine was based on the responsive shrinkage of the MIP that was triggered by the target binding. The MIP gel-SPR sensor showed a linear response in the range of 9.0 × 10^−12^ to 5.0 × 10^−10^ mol/L (R^2^ = 0.9998); high selectivity to the analyte compared to its structurally related analogues and the LoD for the analyte was 0.471 ng/L for tap water and 0.772 ng/kg for soil [86], while, soft, deformable, MIP NPs of about 50 nm diameter were coupled to a plastic optical fiber to detect the protein biomarker, human transferrin (HTR). The MIP NPs were observed to deform at binding to HTR, with the mean Young’s modulus measured by AFM passing from 17 ± 6 kPa for free NPs to 56 ± 18 kPa for bound NPs. Such analyte-induced MIP-NP-deformations amplified the resonance shift, enabling the achievement of ultra-low sensitivities: a LoD of 1.2 fM and a linear dynamic range of concentrations from 1.2 fM to 1.8 pM [41]. To date, it appears that gaining the spatial control of MIP deformation events, and playing with MIP films and MIP NPs that deform close to the plasmon, is the key to dramatically improve the sensor sensitivity, enabling nanosensors characterized by quasi single molecule detection to be attained.

**Table 1 sensors-20-05069-t001:** Summary and comparison of SPR-MIP sensors from Section 4.1.1, Section 4.1.2 and Section 4.1.5 classified on the basis of the MIP functionalization.

Configuration	Preparation of the MIP	MIP Thickness	Analyte	LoD	MIP/Analyte (K_D_)	Sensitivity	Reference
Kretschmann	Spin coating	<nm	δ-Opioid G-protein	-	410 fM	-	[57]
Plastic optical fibre	Spin coating	µm	L-Nicotine	10^−4^ M	0.67 μM	1.3 × 10^4^ nm/M	[58]
Optical fibre	Spin coating	µm	Profenofos	2.5 × 10^−6^ µg/L	-	12.7 nm/log @ 10^−4^ µg/L	[59]
Kretschmann	Photografting	40 nm	Domoic acid	5 μg/L	EC_50_ 58 μg/L	-	[61]
Kretschmann	Photografting	60 nm	Testosterone	10^−12^ M	-	-	[62]
Kretschmann	Photografting/RAFT	nm	17β-Estradiol	1.15 × 10^−15^ M	-	-	[63]
Kretschmann	Photografting/RAFT	nm	Progesterone	0.3 × 10^−19^ M	-	-	[64]
Kretschmann	Photografting/ATPR	nm	Ovalbumin	6.4 ng/mL	-	-	[65]
Kretschmann	Photografting/iniferter	nm	Bovine serum albumin	5.6 × 10^−9^ M	0.17 × 10^−8^ M	7.4 µRIU	[66]
LSPR	Au NPs embedded in MIP	6 µm	Dopamine	10^−6^ M	-	-	[77]
LSPR	MIP NPs & Au NPs	nm	Bisphenol A	<10^−9^ M	4.7 × 10^−9^ M	-	[80]
LSPR	Au-NPs hybrid MIP-microgels	µm	Glucose	0.6 × 10^−3^ M	-	-	[81]
LSPR	Fe_3_O_4_@polydopamine NPs	-	Chlorpyrifos	0.76 × 10^−9^ M	-	-	[82]
LSPR	Au-nanostars embedded in MIP	µm	Trinitrotoluene	2.4 × 10^−6^ M	10^−5^ M	8.5 × 10^4^ nm/M	[83]
LSPR	Ag@AuNPs hexagonal boron nitride (HBN) nanosheets and MIP	34 nm	Etoposide	0.4 × 10^−12^ M	-	-	[84]
Kretschmann	Swellable MIP spin coated	µm	Theophylline	10^−6^ M	-	-	[85]
Kretschmann	Swellable MIP spin coated	-	Dichlorobenzidine	9 × 10^−9^ M	10^−12^ M	-	[86]
Plastic optical fibre	Swellable MIP NPs grafted	10–50 nm	Serum transferrin	1.2 × 10^−15^ M	10^−15^ M	-	[41]

### 4.2. MIP and Waveguides

The simplest photonic structure to use in biosensing is a waveguide. In a waveguide, the light is physically confined by the different refractive index between the waveguide material and the surrounding and each event occurring at the waveguide interface produces a variation in the local refractive index, and consequently, a variation in the light spectrum, as described in paragraph 3. Different materials, both organics and inorganics, have been used for waveguides fabrication and various are the measurement configurations reported (see Table 2). Among the different optical configurations, the following are coupled to MIPs and herein discussed: “free-standing filaments” that are obtained by polymerizing MIP structures in a filamentous shape, after the polymerization the MIP filament is peeled off from the substrate and used as free-standing. The integrated optical waveguide (IOW) uses the longitudinal total internal reflection of a light beam through a planar waveguide. The interferometer, Young interferometer or Fabry-Peròt interferometer represent different configurations that take advantage of interfering waves. Diffraction gratings are optical components with a periodic structure that splits and diffracts light into several beams travelling in different directions. Optical fibers are cylindrical dielectric waveguides, surrounded by a cladding layer that transmits light along the longitudinal axis by total internal reflection. Finally, lossy mode resonance structures combine the guided mode in the core of the optical fiber and the lossy mode in semiconductor metal oxide layer.

A first attempt at coupling the MIP and waveguide is reported in 2001. Yan and Kapua fabricated an MIP microstructure on a silicon wafer using micromolding in capillaries: through a polydimethylsiloxane (PDMS) stamp, they built MIP micromonoliths [87]. In this configuration, the MIP was itself the waveguide, and the specific interaction of the analyte 2,4-dichlorophenoxyacetic (2,4-D) was guaranteed by the imprinted cavities. Following this idea, Brazier and coworkers patterned polymer waveguides onto a silicon substrate, imprinting polyurethane filaments with anthracene [88]. Although they did not reach high performances, they laid the groundwork for the future works. Two years later, the same authors published a paper with a detailed study of the optical properties revealing that MIP parameters have to be optimized, although the fabrication technique based on micromolding capillaries allowed the achievement of the preparation of MIP arrays on a single chip, enabling the simultaneous analysis of multiple analytes [89].

A different technique was employed by Walker and coworkers [90]. A submicron molecularly imprinted sol-gel film was deposited onto a waveguide made by a SiO_2_/TiO_2_ layer on microscope glass to specifically detect the explosive 2,4,6-trinitrotoluene (TNT). When the analyte binds to the MIP film, it is converted into a colored anionic form facilitated by an amine group remaining in the binding site after cleavage of the template. The selectivity is, therefore, assured by the specific binding to the imprinted sites and by the chemistry required to form the colored product. A LoD of 5 ppb was achieved.

A sol-gel technique was also used by Edminton and coworkers to deposit an MIP layer onto a Si_3_N_4_/SiO_2_ waveguide for TNT detection [91]. The sensor was based on an interferometric configuration, where two waveguides (the reference and the sensing one, respectively) were crossed by laser light. The two waveguides produced an interference useful for the detection of wavelength shift. The configuration allowed three orders of magnitude to be gained with respect to the work of Walker et al., and there was a good selectivity for TNT compared to structural analogues. The MIP layer was about 10 times thinner than that proposed by Walker and colleagues, which undoubtedly represents a great advantage (Table 2).

Barrios and coworkers [92] determined the basic optical properties (refractive index, optical attenuation and optical response to the recognition of the antibiotic enrofloxacin) of their MIP film, depositing it onto a glass slide in a grating shape or as a bulk film. The optical parameters were then used to simulate the geometry of a Si_3_N_4_/SiO_2_ waveguide in order to have the highest performance. To achieve this result, two effects produced by the analyte binding to the MIP were considered: a variation in the film refractive index and a variation in the film thickness. To take into account the first effect, they hypothesized an infinite film thickness, while for the second one, they imposed a 300 nm MIP thickness value. The parameters that better match fabrication costs with sensor performances resulted in a waveguide width of ~1 μm and a waveguide thickness between 60 and 80 nm (in quasi-transverse-magnetic (TM) operational mode).

A polymeric integrated Young interferometer functionalized with an MIP film for melamine detection was instead proposed by Aikio and coworkers [93]. An input waveguide branched into four waveguides, forming two parallel Young interferometers integrated into one sensor chip. Both of the interferometers contained a reference and a measurement waveguide. An MIP or control nonimprinted polymer (NIP) were deposited by spin-coating technique: the MIP interferometer was meant to measure the specific binding of analyte, while the NIP interferometer measured the nonspecific one. The phase change, due to the binding of melamine to MIP film, was four times higher with respect to the corresponding NIP film, and a saturation above 0.5 g/L was recorded.

A different waveguide configuration is represented by an optical fiber. In a fiber, the inside radiation pathway is determined by the total internal reflection, and consequently, fibers act as a waveguide.

The simplest way to functionalize an optical fiber with MIP is dipping the fiber tip into partially polymerized solution. This approach was firstly reported by Jenkins and coworkers in 2001 [94]. They directly dip coated the distal end of a glass fiber into a prepolymerized solution for the specific detection of organophosphate pesticides and insecticides, with detection limits less than 10 parts per trillion (ppt), with an extended linear dynamic range (ppt to ppm) and response times of less than 15 min. Another work was reported by Queiros and coworkers [95] where a glass fiber tip was dip coated in the MIP mixture and then polymerized for the specific recognition of Microcystin-LR, an hepatotoxin present in drinking-water. The comparison between MIP and NIP membrane revealed that MIP showed better performances than NIP and also a higher stability to temperature. In a similar way, Xiong and coworkers dipped an optical fiber into an MIP solution designed for Bisphenol A recognition, finally inserting the prepared optical fiber into a transparent capillary for the measurements [96], taking advantage of the evanescent wave field produced on the fiber core surface to excite the bisphenol A fluorescence. The LoD was 1.7 × 10^−9^ g·mL^−1^; selectivity and stability were proved.

Similar to the approach followed by Walker and coworkers [90], Nguyen et al. immobilized a fluorescent MIP to the distal end of an optical fiber for cocaine detection [97]. The carboxylic groups on the fluorophore inside the MIP cavity and the amine group on the analyte formed a complex that enhanced the fluorescence signal at the analyte binding. The system response to cocaine was in the concentration range 25–500 μM. Instead, Ton and coworkers [98] coated a polystyrene optical fiber with a fluorescent microMIP, containing N-(2-(6-4-methylpiperazin-1-yl)-1,3-dioxo-1H-benzo[de]isoquinolin-2(3H)-yl-ethyl)acrylamide [99], and demonstrated the sensor ability to detect the herbicide (2,4-D). The evanescent wave passing through the fiber excited the MIP. The analyte binding to the MIP produced a strong fluorescence enhancement; the fluorescence intensity was proportional to the analyte concentration in the nanomolar range, with a Limit of Quantification (LoQ) of 2.5 nM. Moreover, the authors proposed a further advancement [98]: the evanescent wave passing through the polystyrene optical fiber was exploited to photopolymerize the fluorescent MIP directly on the fiber, and the performance of the in situ fabricated MIP-sensor was compared to the coated sensor, showing similar performances.

Taking advantage of the lossy mode resonance, Usha and Gupta [100] built a nanocomposite of ZnO/MoS_2_ on a silica core fiber optic immobilizing an MIP selectively designed for p-cresol, a highly toxic phenolic compound detected in urine. The nanocomposite was designed to enhance the refractive index of the system resulting in a low detection limit for the analyte; indeed, the reached LoD was of 28 nM.

A more complicated configuration was adopted by Carrasco et al. [101]: an optical fiber was etched creating microwells of approximately 3.1 μm in depth, and MIP (or NIP) fluorescently-encoded microspheres were deposited inside the wells and imaged through epifluorescence microscopy. The specific detection of the analyte enrofloxacin, a fluoroquinolone antibiotic widely used for both human and veterinary applications, was proved in a competitive assay: a fluorescent analogue of the antibiotic competes for active sites within the imprinted microspheres, highlighting the high specificity of MIP. Interfering substances were tested to assess cross-reactivity, which was negligible. Moreover, tested with sheep serum samples (diluted 6-fold), the results suggested that matrix effects were negligible. The excellent long-term stability of these materials was demonstrated for over five years of experiments.

In a recent and extensive review presented by Gauglitz [52] on optical detection systems based on fibers and waveguides, MIPs are acknowledged as a valid alternative to antibodies in complex matrices and real samples, combining good specificity, short time of responses and selectivity.

**Table 2 sensors-20-05069-t002:** MIPs coupled to waveguides.

Waveguide Material	Optical Configuration	MIP Thickness	Analyte	LoD or LoQ	Reference
Polymer	Free-standing filaments	20 µm	2,4-Dichlorophenoxy acetic (2,4-D)	LoQ: 0.021 µmol/g	[87]
Polymer	Free-standing filaments	<20 µm	Anthracene	LoQ: 1.6 µmol/g	[88]
Polymer	Free-standing filaments	100 µm	Anthracene	-	[89]
SiO_2_/TiO_2_	IOW	∼0.3–1.0 µm	2,4,6-Trinitrotoluene (TNT)	LoD: 5 ppb	[90]
Si_3_N_4_/SiO_2_	Interferometry	20–120 nm	TNT	LoD: 2.4 ppt	[91]
Si_3_N_4_/SiO_2_	Diffraction grating	322 nm	Enrofloxacin	LoQ: <50 µM	[92]
Polymer	Young interferometer	-	Melanine	LoQ: <0.1 g/L	[93]
Glass	Fabry-Pérot interferometer	19 µm	Microcystin-LR	LoQ: >1.8 μg/L	[95]
Glass	Fiber microarrays	3.1 µm	Enrofloxacin	LoD: 40 nM	[101]
Glass	Fiber	200 µm	Organophosphates	LoD: <10 ppt	[94]
Glass	Fiber	<5 µm	Bisphenol A	LoD: 1.7 ng/mL	[96]
Glass	Fiber	-	Cocaine	LoQ: <25 μM	[97]
Polymer	Fiber coupled to spectrofluorimeter	few µm to nm	2,4-D	LoQ: 2.5 nM	[98]
ZnO/MoS_2_	Lossy mode resonance	1.2 µm	p-Cresol	LoD: 28 nM	[100]

### 4.3. MIP and Photonic Crystals

An alternative approach for developing optical biosensors is based on Photonic Crystals (PC) structures. These systems consist of two or more materials with different refractive indices, which are periodically arranged in space and manipulate light in the scale of optical wavelengths. From a historical point of view, the PC, where proposed in the pioneering work of Yablonovitch [102] and John [103], aimed to create materials that can manipulate photons in a way that resembles a semiconductor crystal and influences the properties of electrons. Indeed, the shifts in the reflection peak of these 3D photonic structures and the high surface to volume ratio makes them interesting for sensing [104]. The detection systems for PCs ranges from fluorescence [105], to enzymatic reactions [106], to label-free, just by observing the reflection peak change of the structures [107]. Moreover, both inorganic and hydrogel-based 3D photonic structures have been reported.

PC configurations range from 1D structures, based on multilayers Bragg stack films, to 3D ones, such as opal systems (both direct or inverse). The most exploited configurations are the opals due to their low realization cost, flexibility, lightweight and power free employment. Moreover, these systems are suitable for the analysis of both liquids and gasses.

The inverse opal configuration is, to date, the most chosen for the development of MIP PC sensors [108] (see Table 3). Different opal 3D structures have been reviewed: the main configuration is based on silica nanoparticles covered with MIPs deposited onto a solid substrate “film” (Table 3), while another configuration is based on MIP photonic hydrogel particles based on an inverse opal structure and called “hydrogel”, and the last configuration relates to silica microspheres decorated with MIP “microspheres”. Dai and coworkers [109] combined the features of an inverse opal colloidal crystal and an MIP hydrogel, composed of acrylamide and EGDMA, to detect 2-butoxyethanol (2BE), a pollutant associated with hydraulic fracturing contamination, with an LoD of 3.4 ppb. Reflection measurements allowed the correlation of the Bragg’s peak position shifts of MIP PC with the 2BE concentrations from 1 ppb to 100 ppm. A similar approach was developed for the recognition of pesticides, in particular for the detection of parathion. Zhang and colleagues used a gold doped inverse opal PC and an MIP hydrogel (MAA, NIPAM, EGDMA) [110]. A shift in wavelength (Δ*λ*) of the reflection peak position of about 10 nm was reported for a concentration of parathion of about 10 ng/mL.

More recently, Huang et al. [111] developed a portable label-free inverse opal photonic sensor, constituted of MIP hydrogel particles, addressed to pesticides, such as methanephosphonic acid (MPA), by means of colorimetric monitoring. Authors proved that a concentration of MPA of 10^−6^ M induced a large red shift of the Bragg peak that can be seen to the naked eye, with a good recovery over 5 cycles.

The sensor reported by Hou and coworkers [112] consists of an inverse colorimetric PC sensor (see Figure 3) based on hydrophilic–hydrophobic patterned MIP-PC for the detection of tetracycline. The main advantage of this strategy was to realize MIP structures of millimetric dimension on the hydrophobic PC substrate. Upon analyte binding, the MIP-PC sensor underwent a colorimetric transition from cyan to red (optical shift > 200 nm), clearly visible to the naked eye. Moreover, varying the diameter of the MIP dots permits the variation of the detection range of the sensor. Overall, the sensitivity of this MIP-PC sensor was superior by one order of magnitude with respect to that of PC based on MIP films [113]; the LoD was 2 nM.

Sai and coworkers [114] reported on an MIP-PC sensor with improved sensitivity for chloramphenicol (Cm). The system exploited the combination of the features of opals (colloidal crystals) and peculiar wettability of different enrichment processes of arrays of MIP spheres embedded in the PDMS matrix. The authors reported a sensitivity of about 1.5 nM for Cm with a linear response in the range of 1 to 10,000 nM. More recently, Chen et al. [115] developed a smart sensor for the rapid and label-free detection of benzocaine. In fact, authors combined an inverse PC structure with an MIP hydrogel showing high sensitivity and specificity, a fast response time, a regeneration ability and a satisfactory accuracy with respect to high-performance liquid chromatography measurements, with an LoD of about 0.1 mM.

Finally, MIP-PCs have been successfully employed as low cost and optical readout systems in the field of biomolecules. In fact, Kadhem et al. [116] using an inverse hydrogel structure with polyacrylamide as the matrix were able to realize an optical sensor sensitive to 5–100 ppb of testosterone. More recently, MIP-PCs have been used to create protein sensors; in fact, Dabrowski et al. [117], taking advantage of the properties of a semicovalently imprinted inverse opal polythiophene film, reached an LoD of about 13 fM in the detection of human serum albumin, while Chen et al. [118] used surface imprinted silica colloids for hemoglobin bovine (Hb) recognition, demonstrating a sensitivity of 24.8 × 10^−7^ mol/g.

### 4.4. MIP and Whispering Gallery Modes Resonators

Another interesting photonic structure suitable to devise sensors is represented by whispering gallery modes (WGM) resonators. Specificity is given to the WGM resonator by surface functionalization; however, the functional layer is required to be in the range 10–100 nm, in order to remain in the evanescent penetration depth and should be homogeneous in order to preserve the high quality of the transducer. The first applications of WGM resonators in biosensors are reported in 2002–2003 [119,120]. Since then, many works have been published in this field [44,50,121], but only recently, the coupling between the WGM resonator and MIPs has been reported. Just three works related to microring resonators have been published, but all highlight the high sensitivity reached using these systems; for a comparison, see Table 4.

Chen and coworkers [122] developed a biosensor based on a microring resonator functionalized with an MIP for the specific recognition of testosterone. Sensor chips employing grating couplers were manufactured on silicon-on-insulator (SOI) wafers with a 220 nm-thick top silicon layer and 2 µm-thick buried oxide layer. The radius of the ring was 32 µm, and the whole sensor was coated by SiO_2_ with a 2 μm-thick upper cladding layer, while the microring resonator was exposed to the analyzed sample by removing the upper cladding layer in the sensing window. The picture of the optical system used is reported in Figure 4a. The reached LoD was 48.7 pg/mL.

A further example was reported by Xie et al. [123]. A cascaded-microring resonators system was designed for the recognition of progesterone (Figure 4b). The specificity was guaranteed by an MIP, layer which allowed progesterone recognition with high specificity and an LoD of 83.5 fg/mL.

The last reported example applies to the detection of TNT [124]. The microring resonator array chip consisted of a waveguide connected to a fiber-coupled coherent light source that was split into 4 parallel branches with microrings, as shown in Figure 4c. The transmission of each branch could be measured independently. The possibility to measure four rings at the same time allowed for a simultaneous comparison of significant signals; the authors, in fact, compared signals coming from the first ring functionalized with MIP specific for TNT, with the second one functionalized with a NIP, whereby the other rings were deputed to monitor temperature fluctuations. The sensitivity to TNT was about one order of magnitude higher in comparison to other structural analogues.

Due to the high sensitivity of the WGM resonators and the high specificity of MIP functionalization, it is reasonable to think that in the coming years, there will be an increase in the number of examples combining these two systems. The possibility to build thin MIP film or nanoMIP allows for better remaining inside the evanescent tail, maximizing in this way the interaction with the analyte. There is an only one example that reports the MIP functionalization of a microsphere resonator [125]: in this work, a silica sol-gel film was imprinted with a small fluorescent dye (fluorescein isothiocyanate), dipping the microsphere into the solution. The coating technique, the time aging and the extraction procedure were studied finding the best parameters to preserve high quality factor of the resonator. A manual coating for 45–50 s followed by an aging for three days resulted in the best coating procedure, while an oxygen plasma treatment appeared as the best template extraction method. Nevertheless, due to the fragile nature of this kind of WGM resonators, it is more realistic to rely on an increase in MIP functionalization of rings, toroids or other photonic structures more easily built on a photonic chip.

## 5. Conclusions and Perspectives

The time has come to take MIP optical sensing to the next level, that is, the transition from lab-devices to the fabrication of sensing platforms and to their in-field applications. Such a transition has been enabled by the advancements in MIP synthetic strategies and in MIP coupling, which nowadays have been counting on techniques and strategies to precisely control the functionalization of the optical structures. Therefore, the field of MIP optical sensors, currently expanding to photonic crystals, optical waveguides and resonators and the results herein discussed, hold the promises for new generations of MIP optical sensing devices, robust, versatile, sensitive and selective, to face any in field application as well as for the forthcoming analytical challenges.

## Figures and Tables

**Figure 1 sensors-20-05069-f001:**
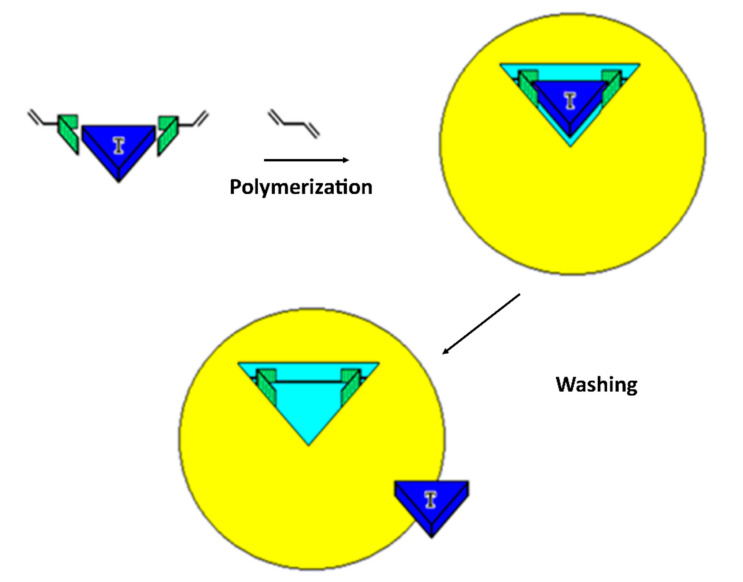
The principle of molecularly imprinted polymer (MIP) synthesis: a given molecular target (blue) is used as a template and placed in solution with monomers (green) with chemical functionalities capable of forming either weak noncovalent interactions, or reversible covalent interactions, with the template. Thermodynamics drive the formation of a complex between the template and the monomers, which after the addition of a crosslinker and of a catalyst, is subsequently “frozen” by polymerization (yellow). The formed MIP material (yellow) bears molecular cavities that present chemo- and stereocomplementarity towards the template. The process of template removal, usually performed by means of washings, leaves binding cavities on the MIP that have affinity and selectivity for their target template and can be used for its selective capture. Adapted from [9].

**Figure 2 sensors-20-05069-f002:**
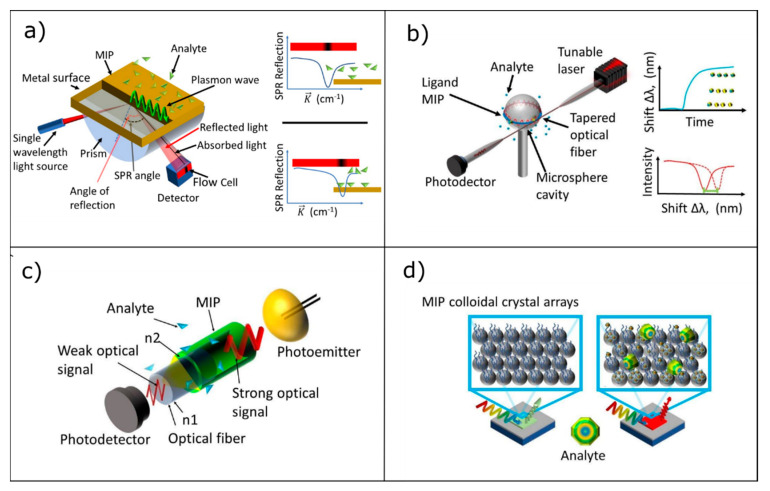
(**a**) Scheme of the surface plasmon resonance (SPR) biosensor where a shift in the surface plasmon (SP) resonance is reported after analyte immobilization, adapted from [45]. (**b**) Sketch of the whispering gallery mode (WGM) biosensor working principle where a resonance shift associated with molecular binding is observed, adapted from [46]. (**c**) Concept of the Optical Waveguide Lightmode Spectroscopy approach based on evanescent wave for biomolecules detection through optical fiber or waveguide taking advantage of different refractive indexes (n1 and n2), adapted from [47]. (**d**) Chromatic sensor exploiting colloidal crystals properties, capable of detecting analytes by observing color changes in the dry condition, adapted from [48].

**Figure 3 sensors-20-05069-f003:**
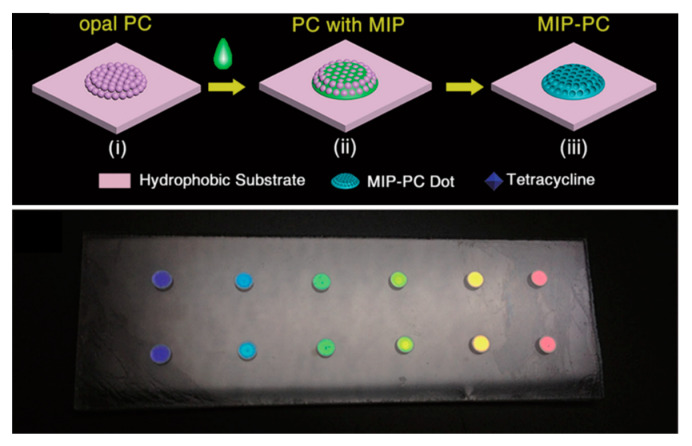
Upper panel: Fabrication process of the MIP–PC. MIP dots of diameter of 1.35 mm are polymerized on hydrophobic substrate. A 10 μL volume of sample solution was dipped on the imprinted PC dot. In the presence of the analyte (tetracyclin), a visible change in color of the dot was visible after 10 min. Lower panel: Photographic image of the PC sensor with different colors of PC dots. Adapted from [112] with permission.

**Figure 4 sensors-20-05069-f004:**
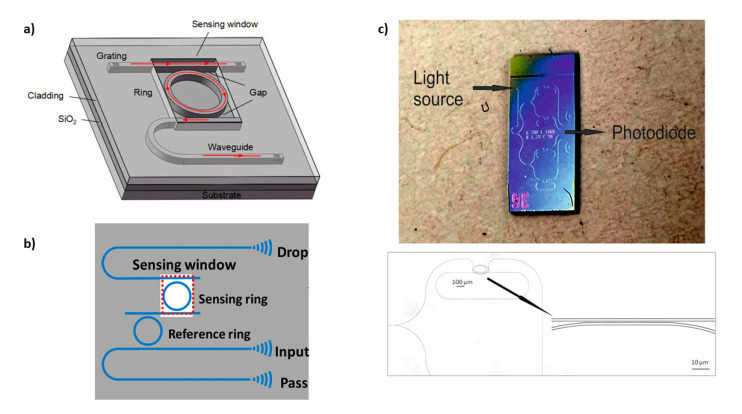
WGM resonator devices with an MIP functional layer. (**a**) Single microring designed by Chen et al. [122]. (**b**) Cascade microring system utilized by Xie et al. [123] with permission from Optics Express. (**c**) Microring resonator array reported by Eisner et al. [124].

**Table 3 sensors-20-05069-t003:** MIPs coupled to photonic crystals.

Photonic Crystal Configuration	Analyte	LoD or LoQ	Reference
Film	2-Butoxyethanol (2BE)	LoD: 3.4 ppb	[109]
Film	Parathion	LoQ: <0.01 ng/mL	[110]
Hydrogel	Methanephosphonic acid (MPA)	LoD: <1.0 µM	[111]
Film	Tetracycline	LoD: <2 nM	[112]
Film	Tetracycline	LoQ: <80 nM	[113]
Film	Chloramphenicol (Cm)	LoD: 1.5 nM	[114]
Film	Benzocaine	LoD: 0.1 mM	[115]
Film	Testosterone	LoD: 4.2 ppb	[116]
Film	Human Serum Albumin (HSA)	LoD: 13 fM	[117]
Microspheres	Hemoglobin bovine (Hb)	LoQ: <0.1 mg/mL	[118]

**Table 4 sensors-20-05069-t004:** MIPs coupled to whispering gallery modes resonators.

WGM Configuration	Analyte	LoD or LoQ	Reference
Microring	Testosterone	LoD: 48.7 pg/mL	[122]
Microring	Progesterone	LoD: 83.5 fg/mL	[123]
Microring	TNT	LoQ: 5 ppb	[124]
Microsphere	Fluorescein	-	[125]

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
