# Peer review of "Molecular Imprinted Polymers Coupled to Photonic Structures in Biosensors: The State of Art"

_sensors, 2020, doi:10.3390/s20185069_

Round 1

Reviewer 1 Report

The paper review concerns the application of molecularly imprinted polymers (MIPs) with optical sensors, mainly focusing on four photonic structures.

Although the paper is relevant to the journal and promise to offer a useful review for the scientific community, the following changes and improvements are needed prior to the publication:

  1. The introduction should be more accurate and supported by more references:
    • examples of the commercially available remote and miniaturised optical sensor should be provided (lines 27-28).
    • Line 46, please indicate the temperature, pH ranges as well as the MIP/sensor sterilisation temperature or techniques.
    • Lines 44-53, please add references.
  2. Overall, a more accurate discussion over selectivity and specificity of the presented senors should be provided in the manuscript. Particular focus should be posed to the MIPs specificity in detecting the target analyte.
  3. Covalent imprinting should be discussed or at least mentioned in the manuscript.
  4. Please improve the quality and accuracy of figure 1. 
  5. Section 3: please enhance the clarity in describing the photonic structures. For instance, the principle of the photonic structures should be briefly stated before discussing examples, pros and cons. Also, all the elements of the equations should be presented.
  6. Section 4: please use tables for summarising the literature and using the main text to discuss the reported works.
  7. It would be better to briefly explain the chemistry and deposition techniques and then describe examples from the scientific literature. Authors may add more schematics and/or tables in sections 4.1.1, 4.1.2, 4.1.3, 4.1.4, 4.1.5, 4.2.
    to make the paper more accessible to the readers.
  8. Please make the subscription consistent (line 211).
  9. Line 225, please add library search results.
  10. Lines 341-342, please provide the values of the previously reported affinity constant vs the one achieved within the cited work [57].
  11. Format of table 1 should be revised.
  12. Resolution of the figures should be improved
  13. Minor repetitions are present throughout the manuscript and should be removed.
  14. English style should be improved.

Author Response

Comments and Suggestions for Authors

The paper review concerns the application of molecularly imprinted polymers (MIPs) with optical sensors, mainly focusing on four photonic structures.

Although the paper is relevant to the journal and promise to offer a useful review for the scientific community, the following changes and improvements are needed prior to the publication:

  1. The introduction should be more accurate and supported by more references:
  • examples of the commercially available remote and miniaturised optical sensor should be provided (lines 27-28).

The text is added of references.

  • Line 46, please indicate the temperature, pH ranges as well as the MIP/sensor sterilisation temperature or techniques.

Thank you for the suggestion. The information is added to the text and references are included.

  • Lines 44-53, please add references.

The text is added of references.

  1. Overall, a more accurate discussion over selectivity and specificity of the presented senors should be provided in the manuscript. Particular focus should be posed to the MIPs specificity in detecting the target analyte.

More discussion on the specificity and selectivity is added to the introduction and in the sections discussing the MIPs coupled to the different optical configurations.

  1. Covalent imprinting should be discussed or at least mentioned in the manuscript.

The covalent imprinting and the related references are added to the manuscript.

  1. Please improve the quality and accuracy of figure 1. 

The quality, the accuracy and the caption to Figure 1 have been improved.

  1. Section 3: please enhance the clarity in describing the photonic structures. For instance, the principle of the photonic structures should be briefly stated before discussing examples, pros and cons. Also, all the elements of the equations should be presented.

Section 3 has been improved, the principles are described with more clarity, the equations are explained.

  1. Section 4: please use tables for summarising the literature and using the main text to discuss the reported works.

More Tables are added, so to summarize the results of the different approaches and permit a comparison.

  1. It would be better to briefly explain the chemistry and deposition techniques and then describe examples from the scientific literature. Authors may add more schematics and/or tables in sections 4.1.1, 4.1.2, 4.1.3, 4.1.4, 4.1.5, 4.2. to make the paper more accessible to the readers.

Concerning the types of photopolymerizations, it would be quite long to introduce the readers to the subject. To help readers, a review marked as [25] is added to the introduction, so to permit the readers to find exhaustive information on the subject.

A Table is added to the Section 4.1 to enable the comparison of sensor performances and characteristics.

  1. Please make the subscription consistent (line 211).

This is an editing problem: in the uploaded word file the subscription is consistent, but not in the pdf. We will make sure that it will be consistent in the proofs.

  1. Line 225, please add library search results.

The result of the literature search and the source of the search are now specified in the text.

  1. Lines 341-342, please provide the values of the previously reported affinity constant vs the one achieved within the cited work [57].

Added in the text and Kd is also added to the summarizing Table 1.

  1. Format of table 1 should be revised.

Table, that now is n.2, has a revised format.

  1. Resolution of the figures should be improved

The resolution of the Figure has been improved.

  1. Minor repetitions are present throughout the manuscript and should be removed.

Thank you, repetition have been removed.

  1. English style should be improved.

We corrected the language.

Reviewer 2 Report

The review is very complete with sufficient references and well structured. However, there are small details, which will be discussed later, that should be taken into account before publication.

Figure 1 should be completed by indicating the components in the picture itself as it would help in its understanding. In addition, in the figure caption, it is indicated that the interactions are non-covalent. Although in this almost if they are, it is implied that this is always the case. Therefore, I recommend the authors to review it.

At the beginning of section 3, the authors should introduce a short paragraph that would introduce this section since they simply start to name them.

Figure 2 should be improved as it looks very small in some of them. These are original images? no references are included.

In section 4 the combination of the MIPs with the previously mentioned photonic structures is commented but in a different order. Is there any reason? wouldn't it be better that both sections name the structures in the same order?

Table 2 cannot be read correctly and makes it difficult to understand.

Author Response

Comments and Suggestions for Authors

The review is very complete with sufficient references and well structured. However, there are small details, which will be discussed later, that should be taken into account before publication.

Figure 1 should be completed by indicating the components in the picture itself as it would help in its understanding. In addition, in the figure caption, it is indicated that the interactions are non-covalent. Although in this almost if they are, it is implied that this is always the case. Therefore, I recommend the authors to review it.

We thank the Reviewer for the comment, the caption to Figure 1 and the paragraph about the molecular imprinting process have been modified so to include covalent imprinting.

At the beginning of section 3, the authors should introduce a short paragraph that would introduce this section since they simply start to name them.

An introductory paragraph is added to section 3.

Figure 2 should be improved as it looks very small in some of them. These are original images? no references are included.

Figures have been modified and improved. Granted permissions, or source are referenced.

In section 4 the combination of the MIPs with the previously mentioned photonic structures is commented but in a different order. Is there any reason? wouldn't it be better that both sections name the structures in the same order?

The same order is maintained throughout the whole manuscript.

Table 2 cannot be read correctly and makes it difficult to understand.

More Tables are added, to allow the reader to compare strategies. The tables in word appears with the correct layout. We will make sure the pdf will keep the layout.

Reviewer 3 Report

This manuscript presents an extensive review of optical biosensors based on polymer substrates.

I recommend to make the following modifications:

  1. The quality of the figures in my copy of the manuscript is low, in particular figures 4-b a 4-c can be hardly viewed. Please verify that the definitive figures have a proper resolution.
  2. Explain the meaning of the yellow draw in figure 1.
  3. The description of figure 3 is quite confusing, please rewrite the following part of figure cation “Contact angles of the MIP–PC 523 dot c) (CA 102.8) and PDMS substrate d) (CA 1153.1). (Inset) The microscope image of the MIP–PC 524 dot in cyan color [89].”
  4. Explain in the text or n the caption of figure 3 what the contact angles are. Explain what are the acronyms CA 102.8 and A 1153.1 to make the discussion consistent.
  5. Use a single stay in all figures, in particular modify the capital letters A), B) and C) in figure 4, and remove (a) in picture B) to avoid confusion.
  6. Rewrite the expression “ that is deputed to the selective interaction with a target analyte” in a different way, I do not consider the word “deputed “ very appropriate in this context.
  7. Rewrite as well the sentence “The time has matured to take MIP optical sensing to a next level”, the adjective “matured” is not very convenient for the name ‘time’.

Author Response

Comments and Suggestions for Authors

This manuscript presents an extensive review of optical biosensors based on polymer substrates.

I recommend to make the following modifications:

  1. The quality of the figures in my copy of the manuscript is low, in particular figures 4-b a 4-c can be hardly viewed. Please verify that the definitive figures have a proper resolution.

We are sorry for the inconvenience. The quality of all the Figures have been improved.

  1. Explain the meaning of the yellow draw in figure 1.

The caption of Figure 1 is modified.

  1. The description of figure 3 is quite confusing, please rewrite the following part of figure cation “Contact angles of the MIP–PC 523 dot c) (CA 102.8) and PDMS substrate d) (CA 1153.1). (Inset) The microscope image of the MIP–PC 524 dot in cyan color [89].”

The text is now simplified and confusing part have been removed or rewritten.

  1. Explain in the text or n the caption of figure 3 what the contact angles are. Explain what are the acronyms CA 102.8 and A 1153.1 to make the discussion consistent.

As before, the Reviewer was right and there was confusion. The contact angles reported in the original manuscript did contain an error (were 10.28 and 115.3 degrees). In the revised form we did not report the values of the contact angles of the original paper, as the information is very specific and there is no space to explain it in details. Instead, we focused on the structure of the MIP sensor. For this Figure 3 has been re-drawn and adjusted.

  1. Use a single stay in all figures, in particular modify the capital letters A), B) and C) in figure 4, and remove (a) in picture B) to avoid confusion.

Letters of Figure 4 and in general of all the Figures are modified to be consisten.

  1. Rewrite the expression “ that is deputed to the selective interaction with a target analyte” in a different way, I do not consider the word “deputed “ very appropriate in this context.

The sentence is now modified.

  1. Rewrite as well the sentence “The time has matured to take MIP optical sensing to a next level”, the adjective “matured” is not very convenient for the name ‘time’.

The sentence is now modified.

Round 2

Reviewer 1 Report

Dear Authors,

Thank you for addressing all my comments. I accept the manuscript in the present form.

Reviewer 3 Report

The manuscript has been modified according to my previous report.